# Public Health Challenges in Saudi Arabia during the COVID-19 Pandemic: A Literature Review

**DOI:** 10.3390/healthcare11121757

**Published:** 2023-06-15

**Authors:** Haytham A. Sheerah, Yasir Almuzaini, Anas Khan

**Affiliations:** 1International Collaborations, Ministry of Health, Riyadh 11176, Saudi Arabia; 2Public Health Authority, Riyadh 13351, Saudi Arabia; almuzaini.yasir1@gmail.com; 3Department of Public Health, College of Public Health, Imam Abdulrahman Bin Faisal University, Dammam 31445, Saudi Arabia; 4Global Center for Mass Gatherings Medicine, Ministry of Health, Riyadh 11176, Saudi Arabia; khanaa@moh.gov.sa; 5Department of Emergency Medicine, College of Medicine, King Saud University, Riyadh 12372, Saudi Arabia

**Keywords:** public health, Saudi Arabia, COVID-19, health regulations, psychological impacts, vaccination, Hajj and Umrah, travel regulations

## Abstract

Similar to most countries, Saudi Arabia faced several challenges during the novel coronavirus disease 2019 (COVID-19) pandemic, some of which were related to the religious position of the country. The main challenges included deficits in knowledge, attitudes, and practices toward COVID-19, the negative psychological impacts of the pandemic on the general population and healthcare workers, vaccine hesitancy, the management of religious mass gatherings (e.g., Hajj and Umrah), and the imposition of travel regulations. In this article, we discuss these challenges based on evidence from studies involving Saudi Arabian populations. We outline the measures through which the Saudi authorities managed to minimize the negative impacts of these challenges in the context of international health regulations and recommendations.

## 1. Introduction

The declaration of the COVID-19 outbreak as a pandemic by the World Health Organization (WHO) on 11 March 2020 prompted countries to take immediate action and implement stringent precautions [1]. As a result, many nations adopted rigorous measures to combat the spread of COVID-19, such as implementing lockdowns, restricting travel, and shutting down schools and universities. Unfortunately, these measures had significant repercussions, including severe economic challenges, a rise in unemployment rates, and heightened levels of stress and anxiety across the globe [2,3,4].

Saudi Arabia encompasses a vast territory of 2.2 million square kilometers and is divided into 13 administrative regions and 118 governorates. It shares borders with several Arab nations. In 2021, the country had a population of 34.1 million, consisting of both native residents and foreigners. Due to its predominantly urban nature, Saudi Arabia boasts numerous residential, commercial, educational, entertainment, and medical facilities. This urban setting makes it susceptible to outbreaks caused by respiratory pathogens [5]. Additionally, the presence of religious sites, which attract millions of pilgrims annually for the Hajj and Umrah pilgrimage, adds to the risk of potential outbreaks.

The first COVID-19 case in Saudi Arabia was detected in a person with recent travel history to an endemic area in Iran on 2 March 2020. The number of COVID-19 cases in the country witnessed a significant surge, rising from 392 cases on 21 March 2020 to 549,518 cases on 22 November 2021. This rapid spread was unexpected and strained the public health system. However, there were some positive developments in terms of daily reported infections. The number of new cases per day decreased from 4757 on 18 June 2020 to 220 on 28 November 2020, further declining to 328 on 28 February 2021, then rising to 1161 on 7 June 2021, before dropping significantly to 39 on 22 November 2021. In parallel with the decline in infections, there was a reduction in the mortality rate due to the pandemic, although the decline was not as substantial. Starting with a single death reported on 24 March 2020, daily deaths increased to 58 on 5 July 2020. The death toll per day remained relatively high at 13 on 28 November 2020, 6 on 28 February 2021, and 15 on 7 June 2021, but eventually decreased to 2 on 22 November 2021. Another important aspect to consider is the daily recoveries. Ever since 12 May 2020, the number of daily recoveries surpassed new cases, although minor fluctuations were observed. The gap between active cases and recoveries that existed during June and July 2020 started to gradually decline from October 2020 and rapidly thereafter [6].

In adherence to international guidelines, Saudi Arabia took decisive actions in the initial stages of the COVID-19 pandemic to curb the transmission of the virus. These measures encompassed implementing lockdowns, enforcing social distancing measures, halting public transportation, closing schools, preventing large religious gatherings, and conducting contact tracing for travelers displaying signs of COVID-19 infection. Furthermore, the government made efforts to enhance various aspects of COVID-19 detection and management capacity. This included the establishment of walk-in respiratory clinics, such as the Tatamman clinic, equipped with trained healthcare professionals and diagnostic equipment across the country, catering to individuals experiencing COVID-19 symptoms. Moreover, Saudi Arabia played a financial role in supporting the provision of COVID-19 vaccines to low-income countries, contributing to global efforts in combating the COVID-19 pandemic [5,6,7,8,9,10,11,12,13,14,15].

Fortunately, before the onset of the COVID-19 pandemic, Saudi authorities had already introduced a transformative economic and social reform plan known as “Vision 2030”. This comprehensive framework played a crucial role in guiding Saudi Arabia’s response to the outbreak. This framework included partnership and coordination, laboratory diagnostics, infection prevention and control, health information management and surveillance, risk communication and community engagement, and vaccination. The country’s strong governance and intersectoral coordination facilitated evidence-based decision-making, resulting in significant advancements in laboratory diagnostics, health information management, and response interventions against COVID-19. Saudi Arabia’s experience in managing the Middle East respiratory syndrome coronavirus (MERS-CoV) and organizing religious mass gatherings also bolstered its preparedness and response capabilities. Collectively, these factors enabled Saudi Arabia to effectively combat the challenges posed by the COVID-19 pandemic [5,6,7,8,9,10,11,12,13,14,15].

Moreover, Saudi Arabia adopted a proactive approach to managing the pandemic by providing both national and international financial support. As an example, during its G20 Presidency, Saudi Arabia committed $150 million to the Coalition for Epidemic Preparedness and Innovations, $150 million to the GAVI Vaccine Alliance, and $90 million to the implementation of the global COVID-19 response [16,17,18]. Additionally, over $100 million was donated to various organizations to aid vaccine development, distribution, and the vaccination needs of lower-income countries.

Nevertheless, Saudi Arabia did face several public health challenges during the COVID-19 pandemic. These challenges encompassed knowledge gaps, attitudes, and practices related to COVID-19, the adverse psychological effects of the pandemic, vaccine hesitancy, management of religious mass gatherings, and the implementation of travel regulations. While some of these challenges were similar to those experienced by other countries, certain issues were unique to Saudi Arabia due to its religious and cultural context. In this context, the manuscript discusses some of these challenges and outlines the measures undertaken by Saudi authorities to address them within the framework of international health regulations.

It is important to note that this discussion does not encompass a systematic review of all relevant publications concerning Saudi Arabia’s experiences during the COVID-19 pandemic. We mainly searched PubMed using the terms COVID-19 and Saudi Arabia, in addition to searching the Saudi official websites of the Ministry of Health and other governmental entities involved in controlling the COVID-19 pandemic. We selected the most featured articles that provided meaningful insights from Saudi populations.

## 2. Public Health Challenges in Saudi Arabia during the COVID-19 Pandemic

### 2.1. Knowledge, Attitudes, and Practices toward COVID-19

The effectiveness of the extensive measures taken by the Saudi Government to combat the COVID-19 pandemic hinged largely on public behavior. The general population needed to adhere to the preventive measures implemented by the government to prevent the spread of the virus. The level of adherence was likely influenced by people’s knowledge and attitudes towards COVID-19. To understand these factors and their underlying influences among individuals in Saudi Arabia, several national studies have been conducted [19,20,21,22,23,24,25,26]. These studies aimed to assess the public’s knowledge and attitudes, as well as the factors that mediate them, providing valuable insights into the mindset of the Saudi population during the COVID-19 pandemic.

The majority of studies examining Saudi individuals’ knowledge, attitudes, perceptions, and practices toward COVID-19 infection, such as an online survey involving 3388 participants, indicated satisfactory levels [19]. However, the same study found that men and younger individuals scored lower in terms of knowledge, attitudes, and practices compared to women and older adults [19]. Another study involving 2393 participants revealed that the majority of individuals were capable (88%) and willing (82%) to self-isolate, but those with lower household incomes had a lower ability and willingness to do so [20]. Similarly, an online survey with 5105 Saudi residents demonstrated that around 90% of participants displayed high knowledge and practice scores regarding hand hygiene and glove and mask usage. However, it also highlighted that men, younger adults, and individuals with lower incomes had lower knowledge and practice scores [21]. Another study covering all Saudi regions and including 443 participants revealed that a significant percentage had awareness and practiced key preventive measures, such as hand-washing, sneezing or coughing into the arm/elbow, avoiding handshakes, maintaining safe distancing, refraining from touching the face, and understanding the importance of social distancing [22]. Participants with lower education exhibited poorer knowledge and practices related to COVID-19 [22]. Furthermore, a study involving 3388 participants showed that socioeconomically privileged individuals had higher levels of knowledge concerning COVID-19 [23].

Similar patterns were observed among healthcare workers. For instance, a study with 597 healthcare workers, including physicians, nurses, medical students, and pharmacists, found that most participants adhered to preventive measures such as hand hygiene, avoiding touching the face, and avoiding crowded places [24]. A nationwide survey of 1040 healthcare workers indicated good knowledge about COVID-19, with better knowledge observed among those with more years of experience and higher education levels [25]. Additionally, a study involving 1226 nursing students from seven Saudi universities demonstrated satisfactory perceptions, knowledge, and preventive behaviors toward COVID-19 [26].

During health crises, there is a high demand for information, and individuals seek information from various sources. Traditional media, including television and newspapers, play a role in disseminating evidence-based information, whereas social media platforms allow individuals to express their opinions, perceptions, and attitudes toward public health policies regarding COVID-19. Individuals may also rely on their social networks, such as family, friends, and coworkers, for information [27,28,29]. As attitudes and practices are closely linked to knowledge, it is important to identify the sources of information on COVID-19 and evaluate their impact on individuals’ attitudes and practices. A study with 3358 participants revealed that most individuals relied on social media and the Saudi Ministry of Health as their primary sources of COVID-19 information. Obtaining information from social media was associated with less optimistic attitudes and lower adherence to preventive measures, whereas those who obtained information from the Saudi Ministry of Health exhibited more optimistic attitudes and greater adherence to preventive measures compared to those who relied on other sources [30].

It should be noted that the studies mentioned above relied on online surveys, which may have been prone to response bias, as individuals with better knowledge, attitudes, and practices are more likely to participate. Furthermore, most of these studies lacked representativeness, making it challenging to generalize the results to the entire Saudi population or healthcare workers in the country. Moreover, these studies identified inequalities, particularly in knowledge, representing a significant public health challenge. Individuals from lower educational and socioeconomic backgrounds were less likely to possess adequate knowledge, emphasizing the need for targeted educational programs to effectively manage the COVID-19 pandemic within the framework of international health regulations.

### 2.2. Psychological Impact of the COVID-19 Pandemic

In addition to the fear of infection, the implementation of interventions such as social distancing, community quarantine, and lockdowns to mitigate the COVID-19 pandemic may have exacerbated its psychological impacts. Several national studies have examined the prevalence and factors contributing to the psychological effects of the COVID-19 pandemic on the general public and healthcare workers in Saudi Arabia. For instance, a study involving 2081 participants found that the prevalence values of depression and anxiety were 9.4% and 7.3%, respectively. Non-Saudi residents, individuals aged 50 years and above, divorcees, retirees, university students, and those with low income were at a higher risk of developing depression, whereas Saudis, married individuals, the unemployed, and those with a high income were more susceptible to anxiety [31]. Another study with 582 undergraduate students reported high levels of depression, anxiety, insomnia, perceived stress, and low levels of resilience during the pandemic. Lower resilience, a high prevalence of insomnia, pre-existing mental health conditions, and learning difficulties were significantly associated with increased levels of depression, anxiety, and stress [32]. Among 628 women (including 74 pregnant women), 89.2% of the pregnant women expressed concerns about the risk of infection for their unborn babies during hospital delivery, and 94.9% of mothers and pregnant women experienced psychological distress [33]. In a study involving 936 university students, 41.1% exhibited depressive symptoms (mild to moderate: 31.7%; severe to very severe: 9.4%), 26.9% had anxiety symptoms (mild to moderate: 15.8%; severe to very severe: 11.1%), and 22.4% had stress symptoms (mild to moderate: 15.2%; severe to very severe: 7.2%). Factors such as knowing someone infected with COVID-19 or who died due to the disease, spending at least 2 h per day reading or watching pandemic-related news, and lacking emotional support from family, university, and society were associated with a higher likelihood of developing psychological problems [34]. Among 1030 patients with chronic diseases across 13 provinces, 21.5% met the criteria for anxiety, and 19.4% had borderline anxiety. Risk factors for anxiety included female sex, lower education, young age, divorce or death of a spouse, treatment with immunosuppressants, and relying on media as a source of COVID-19 knowledge [35]. Among 492 students, 43.3%, 37.2%, and 30.9% experienced symptoms of depression, anxiety, and stress, respectively. Knowing someone infected with COVID-19, fear, and various coping strategies predicted students’ mental health statuses [36].

Similar to the general public, healthcare workers in Saudi Arabia were significantly impacted by the COVID-19 pandemic. A study involving 426 healthcare workers (48.4% physicians, 24.2% nurses, and 27.4% others) found that 69% experienced depression, 58.9% had anxiety, 55.9% faced stress, and 37.3% had inadequate sleep (less than 6 h per day). Those of the female sex, aged 30 years or younger, working in emergency and night shifts, spending at least 2 h per day reading or watching COVID-19 news, and lacking emotional support from family, society, and the hospital were associated with a higher likelihood of experiencing depression, anxiety, stress, and inadequate sleep [37]. Another study with 1101 healthcare workers (242 physicians, 340 nurses, 310 paramedics, and 209 administrative workers) revealed that stress was linked to turnover intention. Importantly, social support mitigated the relationship between stress and turnover intention [38].

Considering the substantial psychological impacts of COVID-19 on the general public and healthcare workers, the provision of psychological support and counseling has been widely promoted in Saudi Arabia. However, in addition to adhering to international health regulations, it is suggested that interventions should particularly target women, younger individuals, healthcare workers, and those with lower educational and socioeconomic backgrounds.

### 2.3. COVID-19 Vaccine Hesitancy

Achieving herd immunity through vaccination is widely recommended, but vaccine hesitancy among the general population remains a significant issue. Investigating COVID-19 vaccine hesitancy and the factors influencing vaccination acceptance among the Saudi population is crucial for implementing effective vaccination strategies. Several national studies have examined the knowledge, attitudes, perceptions, and hesitancy toward COVID-19 vaccination among the general public and healthcare workers. For instance, a study with 2022 participants from various regions of Saudi Arabia found that 76.0% had satisfactory knowledge, 72.4% had positive attitudes, and 71.3% had positive perceptions regarding the use of COVID-19 vaccines [39]. Another study involving 1599 participants reported an overall vaccine acceptance rate of 79.2%. Key factors motivating individuals to receive the vaccine were confidence in government decisions (54.8%) and a personal sense of responsibility to control the pandemic (48.7%) [40].

In contrast, concerns about insufficient clinical trials (11.4%) and potential undiscovered side effects (11%) were the main drivers of vaccine hesitancy [37]. A descriptive analysis of 531 participants showed that 61.8% were willing to receive the COVID-19 vaccine, with higher vaccine hesitancy observed among women (44.9%), individuals aged 34–49 years (47.9%), married individuals (41.9%), employed individuals (39.7%), those with lower educational levels (40%), and urban residents (40.8%) [41]. The primary reason for vaccine acceptance was personal and community protection, whereas concerns about vaccine safety were the main reasons for hesitancy [41]. In a study involving 3101 participants, 44.7% accepted vaccination against COVID-19, whereas 55.3% expressed hesitancy. Younger individuals, men, and those who had previously received seasonal influenza vaccines were more likely to accept COVID-19 vaccination, with concerns about side effects being the main barriers to acceptance [42]. Another study with 310 participants with chronic diseases reported that 52.0% agreed, 33.5% were unsure, and 14.5% refused to receive the COVID-19 vaccine. The most commonly reported concern was the fear of potential side effects [43]. In a study involving 758 participants, 64% expressed willingness to receive the vaccine, whereas 18.3% showed extreme hesitancy. Factors associated with vaccine acceptance included the source of COVID-19 information, perception of the vaccine’s effectiveness against other variants of the virus, prior influenza vaccination, and the potential requirement of mandatory vaccination for international travel [44]. Among 488 older adults, 43.9% expressed willingness to accept the COVID-19 vaccine. Men, highly educated individuals, and those highly concerned about infection were more likely to accept the vaccine. Adverse side effects (27%) and concerns about safety and efficacy (22.6%) were the most frequently cited reasons for vaccine hesitancy [45].

The lack of information highlighted in many studies underscores the importance of implementing relevant intervention strategies. Ongoing awareness programs targeting the entire population should provide updated information, including vaccination strategies for pregnant women and children. Still, further research is needed to assess whether the COVID-19 vaccination campaign has achieved herd immunity in the country. Additionally, studying the potential side effects of COVID-19 vaccines in comparison to their benefits is necessary.

### 2.4. Religious Mass Gatherings

Annually, Saudi Arabia welcomes over 10 million pilgrims from more than 180 countries who come for Umrah and Hajj. However, the movement of a large number of individuals and the congregation of crowds during Hajj and Umrah pose significant health risks, including the spread of respiratory infections, accidents due to overcrowding, pollution, and various other medical concerns. Saudi Arabia, as a host country, had to address these risks by implementing specific response measures [46,47,48,49,50,51].

In the absence of proper planning and preventive measures, mass gatherings of pilgrims can overwhelm the healthcare system of the host country and have implications for global health preparedness. Before traveling to Saudi Arabia for Hajj and Umrah, pilgrims are advised to seek medical advice regarding potential health risks and carry vaccination certificates, which will be inspected by Saudi authorities at the entry port. Even before the COVID-19 pandemic, the Saudi Arabian Ministry of Health provided a guide for pilgrims on health issues, vaccination requirements, and control measures to ensure their safety and prevent infectious disease outbreaks [46,47,48,49].

The provision of health education to pilgrims regarding both communicable and non-communicable diseases, along with preventive measures for disease transmission, is a key focus for the Hajj and health authorities in Saudi Arabia. This approach is also crucial for global public health and disease control, as it promotes risk reduction and compliance with preventive measures. During Hajj and Umrah, pilgrims play a crucial role in maintaining public health by practicing preventive measures such as personal hygiene, managing coughs and sneezes, frequent handwashing with soap and water or sanitizers, using personal protective equipment such as face masks, and properly disposing of waste in designated bins. Wearing face masks is an affordable and effective method to control pathogen exposure in high-risk environments, reducing the transmission of communicable diseases, including COVID-19, and preventing inhalation of airborne particles [46,47,48,49,50,51,52,53].

Considering that COVID-19 can affect travelers and spread locally and globally, it is essential for countries with significant Muslim populations to promptly implement preventive measures to counteract the transmission of the virus. Furthermore, as many pilgrims are older individuals who are at higher risk of mortality, all stakeholders must collaborate and implement rigorous measures to mitigate the spread of the virus and its associated risks worldwide.

In response to the COVID-19 pandemic, Saudi Arabia implemented decisive measures centered around social distancing to reduce human-to-human transmission during Hajj and Umrah. Eventually, new regulations were introduced to facilitate a gradual and safe return to pre-pandemic conditions. These guidelines included the mandatory use of face masks, temperature monitoring at religious sites, and other social distancing measures. The timeline of regulations for mass gatherings was as follows [54]:On 4 March 2020, the Umrah pilgrimage was suspended.On 8 March 2020, external activities in the Grand Mosque were suspended.On 17 March 2020, prayers in all mosques were suspended.On 21 May 2020, permission was granted to resume prayers in all mosques except for the Grand Mosque.On 22 June 2020, a limited number of individuals were allowed to practice Hajj under stringent preventive measures.On 11 August 2021, numerous individuals were granted permission to practice Umrah under strict preventive measures.

### 2.5. Travel Regulations

Saudi authorities implemented a series of travel regulations in accordance with international health guidelines to prevent the transmission of COVID-19. These measures, outlined below, were enacted at various points in time:In January 2020, the Saudi Ministry of Health established a points-of-entry platform at the Command-and-Control Center, following the recommendations of the WHO Director-General’s Emergency Committee, which recognized the COVID-19 outbreak as a global public health emergency.In February 2020, rapid response teams were activated at the Emergency Operation Center of the Saudi Ministry of Health.Additionally, in February 2020, public health emergency plans were put into action at all entry points, involving active surveillance and an immediate reporting system linked directly to the Saudi Ministry of Health.In the same month, health control centers at entry points were equipped with devices and equipment such as thermal cameras and remote thermometers. Manpower was increased, and personal protective equipment was provided.Task forces and joint teams involving all stakeholders were established in February 2020.Throughout February 2020, mechanisms were continuously implemented to update policies and provide online training to healthcare workers.In March 2020, travel was suspended for both citizens and residents traveling to and from Saudi Arabia.Special protocols were prepared in March 2020 to manage specific categories exempted from the travel suspension, including truck drivers at ground crossings, aircraft crews, and ship crews.Stronger coordination and collaboration with health authorities at neighboring countries’ entry points, as well as communication and coordination with international health authorities, were prioritized in March 2020.Travel restrictions were lifted in September 2020, subject to strict preventive measures such as presenting a negative COVID-19 polymerase chain reaction (PCR) test and undergoing self-isolation, starting from January 2021.In December 2020, all international travel was suspended for a period of two weeks.From March to May 2021, travel to and from specific countries with a high prevalence of COVID-19 or those reporting the presence of new strains was suspended.In May 2021, an institutional quarantine protocol was implemented, excluding individuals who had received accredited COVID-19 vaccines.Additionally, in May 2021, citizens who had been vaccinated against COVID-19 (with two doses or one dose received at least two weeks prior to travel) or those who had recovered from COVID-19 within the past six months were permitted to travel outside Saudi Arabia.Starting from August 2021, tourists were allowed to visit Saudi Arabia under strict preventive measures, including presenting a negative COVID-19 PCR test, a COVID-19 vaccination certificate, and undergoing self-isolation.In September and November 2021, travel to and from specific countries with a high prevalence of COVID-19 or those reporting the presence of new COVID-19 strains was once again suspended.In December 2021, strict preventive measures for citizens were eased.As of February 2022, it became mandatory to present a vaccination certificate indicating receipt of a third dose of the COVID-19 vaccine, along with a negative PCR test, for entering and exiting Saudi Arabia, respectively.Finally, in May 2022, all travel restrictions related to COVID-19 were lifted.

## 3. Limitations

Several limitations should be considered before generalizing the conclusions of this review. First, this is a literature review, not a systematic review; therefore, we used evidence from featured articles, and we did not systematically collect all related articles. Second, the included articles in this review were limited by their cross-sectional design that could not assign causality or temporality, the lack of representativeness due to the non-random methods of recruitment or due to selecting populations with certain occupational or socio-demographic characteristics, the increased risk of nonresponse bias as a result of using online surveys for data collection [55], the limited sample size, and the lack of proper adjustment for potential confounders.

## 4. Conclusions

The earliest responses of the Saudi authorities to the COVID-19 pandemic involved the implementation of travel bans, suspension of religious activities, closure of non-essential shops, enforcement of changes at the workplace, and imposition of curfews. However, the country has faced several challenges, including deficits in the knowledge, attitudes, and practices toward COVID-19, psychological impacts of COVID-19, vaccine hesitancy, planning for religious mass gatherings, and travel regulations. The Saudi authorities managed these problems by providing health education programs, disseminating information on COVID-19 via governmental media and social websites, providing psychological counseling to those at risk of experiencing psychological consequences. Religious mass gatherings were temporarily suspended, followed by a gradual return to these practices under social-distancing precautions. Still, more research is needed to examine the long-term effects of the COVID-19 pandemic on the public health of Saudi Arabia and whether the measures taken could minimize these effects. The preparedness of Saudi Arabia for future pandemics can be fostered by critically appraising the challenges faced during the current COVID-19 pandemic.

## Data Availability

Not applicable.

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
