# Peer review of "Public Health Challenges in Saudi Arabia during the COVID-19 Pandemic: A Literature Review"

_healthcare, 2023, doi:10.3390/healthcare11121757_

Round 1

Reviewer 1 Report

The outbreak of COVID-19 throughout the world was a big challenge for people to overcome this. The high throughput screening and accuracy of data analysis have made this possible. Your review article is an interesting one as it explains the challenges facing health authorities worldwide as regarding deficits in knowledge, attitudes, and practices toward COVID-19, the negative psychological impacts of the pandemic on the general population and healthcare workers, vaccine hesitancy and the imposition of travel regulations.  

I hope in your next research you can study the side effects of COVID- 19 vaccines and is vaccination actually leads to herd immunity.? As there are now a big debate about this issue.

Author Response

Thank you sincerely for providing your valuable feedback. We agree with your perspective, and in response to your insightful input, we have taken the necessary steps to enhance our statement by incorporating a distinct phrase that highlights and acknowledges the ongoing debate.

Reviewer 2 Report

The article submitted for review deals with a very interesting phenomenon, which was the reaction of the health care system and society to Covid in Saudi Arabia. The article enriches the knowledge of the positive and negative phenomena caused by the pandemic. The described solutions and system changes can serve as examples of rapid and effective intervention during the threat of a pandemic.
I have two comments:
There is a lack of a more detailed description of how the papers used in writing the paper were selected. What databases did the authors use, what keywords, if any, were used in the selection, were there papers that were excluded from the analysis for some reason? Such a description would strengthen the methodology of the work.
  The second comment relates to the selection of analysis areas/objects of analysis. The titles are very general - perhaps, for example, in the section on "Psychological impact of the COVID-19 pandemic" it would have been possible to separate out subsections on the reaction of the public and health system workers. Such minor corrections could more accurately show the content discussed. 

Author Response

The article submitted for review deals with a very interesting phenomenon, which was the reaction of the health care system and society to Covid in Saudi Arabia. The article enriches the knowledge of the positive and negative phenomena caused by the pandemic. The described solutions and system changes can serve as examples of rapid and effective intervention during the threat of a pandemic.

I have two comments:

There is a lack of a more detailed description of how the papers used in writing the paper were selected. What databases did the authors use, what keywords, if any, were used in the selection, were there papers that were excluded from the analysis for some reason? Such a description would strengthen the methodology of the work.

Response: Thank you for your comments. The manuscript was a literature review not a systematic review; therefore, we used evidence from the most featured articles. However, we did not systematically collect all related evidence. However, we added a phrase in the introduction section highlighting the methods used in this article (line).

  The second comment relates to the selection of analysis areas/objects of analysis. The titles are very general - perhaps, for example, in the section on "Psychological impact of the COVID-19 pandemic" it would have been possible to separate out subsections on the reaction of the public and health system workers. Such minor corrections could more accurately show the content discussed.

Response: We agree with you that the title is general. Yet, our literature review aimed to describe in brief the challenges that Saudi Arabia has faced during the COVID-19 pandemic. Thus, we were not able to extensively describe every section. Still, we added a limitation section summarizing the limitations of this review article.

Reviewer 3 Report

This paper is not a research paper, but a description of the main findings from a series of studies on Covid-19 in Saudi Arabia. Although interesting facts are offered in this text, it is not a systematic literature review that strives to comprehensively identify, appraise, and synthesize all the relevant studies on the given topic.

I therefore have two suggestions: 

1) In the conclusion, rather than a repetition of the substantive keywords, a critical synthesis of all the studies analyzed, and what conclusions should be drawn from them, should be attempted.

2) If individual measures to cope with the pandemic are listed in the timeline, at least basic data on the course of the pandemic in Saudi Arabia should also be provided, for example in the introductory chapter.

Author Response

This paper is not a research paper, but a description of the main findings from a series of studies on Covid-19 in Saudi Arabia. Although interesting facts are offered in this text, it is not a systematic literature review that strives to comprehensively identify, appraise, and synthesize all the relevant studies on the given topic.

I therefore have two suggestions:

  • In the conclusion, rather than a repetition of the substantive keywords, a critical synthesis of all the studies analyzed, and what conclusions should be drawn from them, should be attempted.

Response: Thank you for the feedback. We added a limitation section before the conclusion to describe the limitations of our review article and the limitations of the included articles that we used to retrieve evidence. We also added phrases to the conclusion section about future perspectives.

2) If individual measures to cope with the pandemic are listed in the timeline, at least basic data on the course of the pandemic in Saudi Arabia should also be provided, for example, in the introductory chapter.

Response: We added a paragraph in the introduction section describing the course of the pandemic in Saudi Arabia.

Reviewer 4 Report

Thank you for this narrative review summarizing public health measures as well as challenges for the health (system) owing to the Covid-19 pandemic in Saudi Arabia. Generally, the paper is well-organized and well-written. Just a few minor remarks for further improvement:

- could the authors describe briefly how they selected the literature included in  their review? I am aware this is not a systematic but a narrative review, but nevertheless the scope of references and search terms should be outlined, especially as some Arabian sources might be involved (presumably) which are not usually accessible for an international readership.

- sometimes the description of main findings from single studies, is a bit superficial and might loose an interesting point. For example, on line 150
, it is written "
In another study of 628 women (including 74 pregnant women), 89.2% expressed concerns regarding the risk of infection for their unborn babies..." - I guess it was 89.2% of the pregnant women?!?

- On line 258, it is written "A study including 1,012 participants from 41 nationalities who attended Umrah in 2019 showed that 76% of pilgrims had been immunized before traveling to Saudi Arabia". While the Umrah was in winter 2019/20 (if I am not mistaken) so while the epidemic has already started in China, I wonder about the relevance of vaccination for Covid-19 related problems (this has not been a vaccination against SARS-Cov2, so please clarify)

- In the brief "Conclusions" section, it would be interesting to add a comparison of effectiveness of the Saudi Arabian measures e.g. related to other countries, perhaps Sweden as a fairly unique "laissez-faire" model, the US, Denmark, or whatever seems interesting, and use e.g. (globally available) data on excess mortaility during the critical period. That is, relate the measures a bit more to the health problem or threat.

Author Response

Thank you for this narrative review summarizing public health measures as well as challenges for the health (system) owing to the Covid-19 pandemic in Saudi Arabia. Generally, the paper is well-organized and well-written. Just a few minor remarks for further improvement:

- could the authors describe briefly how they selected the literature included in their review? I am aware this is not a systematic but a narrative review, but nevertheless the scope of references and search terms should be outlined, especially as some Arabian sources might be involved (presumably) which are not usually accessible for an international readership.

Response: Thank you for your feedback.

Response: We added a section by the end of the introduction showing, in brief, the methods used for retrieving articles. We also added a limitation section highlighting the limitations of the literature review and the studies included in this review.

- sometimes the description of main findings from single studies, is a bit superficial and might loose an interesting point. For example, on line 150

, it is written "In another study of 628 women (including 74 pregnant women), 89.2% expressed concerns regarding the risk of infection for their unborn babies..." - I guess it was 89.2% of the pregnant women?!?

Response: Since we had to cover several topics in one review article, we were not able to go deeper in every study. Yet, we clarified the misunderstanding in this part.

- On line 258, it is written "A study including 1,012 participants from 41 nationalities who attended Umrah in 2019 showed that 76% of pilgrims had been immunized before traveling to Saudi Arabia". While the Umrah was in winter 2019/20 (if I am not mistaken) so while the epidemic has already started in China, I wonder about the relevance of vaccination for Covid-19 related problems (this has not been a vaccination against SARS-Cov2, so please clarify)

Response: We removed this paragraph.

- In the brief "Conclusions" section, it would be interesting to add a comparison of effectiveness of the Saudi Arabian measures e.g. related to other countries, perhaps Sweden as a fairly unique "laissez-faire" model, the US, Denmark, or whatever seems interesting, and use e.g. (globally available) data on excess mortaility during the critical period. That is, relate the measures a bit more to the health problem or threat.

Response: The aim of our review article was to address the public health challenges faced by Saudi Arabia during the COVID-19 pandemic. Comparing the measures taken in Saudi Arabia to those in Western countries might distract readers from the message we aimed to address. Further, the situation in Denmark and Sweden is completely different from that of Saudi Arabia. For example, Saudi Arabia is a major religious destination that attracts more than 10 million pilgrims per year, while Denmark and Sweden are not. There are significant differences in the political systems, the sociodemographic characteristics of the populations, and the geographic site. The Swedish laissez-faire model is fundamentally different from the Saudi one where most measures were almost mandatory. However, we believe that comparing models is very interesting and could provide good insights by comparing the measures and the outcomes across countries. Yet, such comparisons need further research.

Round 2

Reviewer 3 Report

Dear authors, thank you for the revision, I have no more objections.